# Developing a reporting guideline for artificial intelligence-centred diagnostic test accuracy studies: the STARD-AI protocol

Viknesh Sounderajah [1,2] Hutan Ashrafian,[1,2] Robert M Golub,[3] Shravya Shetty,[4] Jeffrey De Fauw,[5] Lotty Hooft,[6,7] Karel Moons,[6,7] Gary Collins [8] David Moher [9] Patrick M Bossuyt,[10] Ara Darzi,[1,2] Alan Karthikesalingam,[11] Alastair K Denniston,[12,13,14,15] Bilal Akhter Mateen [16] Daniel Ting,[17] Darren Treanor,[18] Dominic King,[19] Felix Greaves,[20] Jonathan Godwin,[5] Jonathan Pearson-Stuttard,[21] Leanne Harling,[1] Matthew McInnes,[22] Nader Rifai,[23] Nenad Tomasev,[5] Pasha Normahani,[1] Penny Whiting,[24] Ravi Aggarwal,[1,2] Sebastian Vollmer,[16] Sheraz R Markar [1] Trishan Panch,[25] Xiaoxuan Liu,[12,13,14,15] On behalf of the STARD-AI Steering Committee

For numbered affiliations see end of article.

**Correspondence to**
Hutan Ashrafian;
hutan@imperial.ac.uk

## ABSTRACT

**Introduction** Standards for Reporting of Diagnostic Accuracy Study (STARD) was developed to improve the completeness and transparency of reporting in studies investigating diagnostic test accuracy. However, its current form, STARD 2015 does not address the issues and challenges raised by artificial intelligence (AI)-centred interventions. As such, we propose an AI-specific version of the STARD checklist (STARD-AI), which focuses on the reporting of AI diagnostic test accuracy studies. This paper describes the methods that will be used to develop STARD-AI.

**Methods and analysis** The development of the STARD-AI checklist can be distilled into six stages. (1) A project organisation phase has been undertaken, during which a Project Team and a Steering Committee were established; (2) An item generation process has been completed following a literature review, a patient and public involvement and engagement exercise and an online scoping survey of international experts; (3) A three-round modified Delphi consensus methodology is underway, which will culminate in a teleconference consensus meeting of experts; (4) Thereafter, the Project Team will draft the initial STARD-AI checklist and the accompanying documents; (5) A piloting phase among expert users will be undertaken to identify items which are either unclear or missing. This process, consisting of surveys and semistructured interviews, will contribute towards the explanation and elaboration document and (6) On finalisation of the manuscripts, the group's efforts turn towards an organised dissemination and implementation strategy to maximise end-user adoption.

**Ethics and dissemination** Ethical approval has been granted by the Joint Research Compliance Office at Imperial College London (reference number: 19IC5679). A dissemination strategy will be aimed towards five groups of stakeholders: (1) academia, (2) policy, (3) guidelines and regulation, (4) industry and (5) public and non-specific

### Strengths and limitations of this study

► There are no specific reporting standards for artificial intelligence (AI) diagnostic test accuracy studies.

► We are developing a specific set of reporting standards for AI diagnostic test accuracy studies; Standards for Reporting of Diagnostic Accuracy Studies-AI (STARD-AI).

► This will help key stakeholders to appraise quality and compare diagnostic test accuracy of AI models that are reported in scientific studies.

► STARD-AI will be the product of an extensive evidence generation process that is led by multiple stakeholders (clinician scientists, computer scientists, journal editors, Enhancing Quality and Transparency of Health ResearchNetwork representatives, reporting guideline developers, epidemiologists, statisticians, industry leaders, funders, health policy makers, patients, legal experts and medical ethicists).

► Views of Delphi panellists may differ from those experts who decline participation.

stakeholders. We anticipate that dissemination will take place in Q3 of 2021.

## INTRODUCTION

Artificial intelligence (AI) is commonly cited as an imminent disruptive innovation[1] within the health sector. If used successfully, AI has the potential to tackle (1) the high rate of avoidable medical errors, (2) workflow inefficiencies and (3) delivery inefficiencies associated with modern healthcare provision.[2] The majority of AI interventions that are

close to translation are in the field of medical diagnostics.[3] In the current paradigm, diagnostic investigations require timely interpretation from an expert clinician in order to generate a diagnosis and to subsequently direct episodes of care. However, the recurring issue with the present system is that diagnostic services are inundated with large volumes of work, which often exceeds workforce capacity[4]; COVID-19 being an immediate case in point. In order to address this, diagnostic AI algorithms have positioned themselves as medical devices that may achieve diagnostic accuracy comparable to that of an expert clinician while concurrently alleviating health-resource use. Although this paradigm shift may seem imminent, it is crucial to note that much of the evidence supporting diagnostic algorithms has been disseminated in the absence of AI-specific reporting guidelines. Without this guidance, and in a relatively nascent area, key stakeholders are poorly placed to appraise quality and compare diagnostic accuracy between scientific studies.

The Standards for Reporting of Diagnostic Accuracy Studies (STARD) 2015 statement remains the most widely accepted set of reporting standards for diagnostic test accuracy studies.[5] STARD was developed to improve the completeness and transparency of studies investigating diagnostic test accuracy. It consists of a checklist of 30 items that authors are strongly encouraged to address when reporting their diagnostic test accuracy studies. It is endorsed by over 200 biomedical journals[6] and studies have shown that adherence to the STARD checklist leads to improved reporting of key study parameters.[7 8]

However, in its current iteration, STARD 2015 is not designed to address the issues and challenges raised by AI-driven modalities. Issues include unclear methodological interpretation (eg, data preprocessing steps, model development choices and the use of external validation datasets), the lack of standardised nomenclature (eg, the varying definition of the term 'validation'), as well as the use of unfamiliar outcome measures (eg, Jaccard similarity coefficient and F-score). Until these issues are addressed, achieving comprehensive evaluations of these technologies and their potential translational benefits will remain limited.

In order to tackle these problems, we propose an AI-specific STARD guideline (STARD-AI) that aims to focus on the reporting of AI diagnostic test accuracy studies.[9] This work is complementary to the other AI centred checklists listed in the Enhancing Quality and Transparency of Health Research (EQUATOR) Network programme (www.equator-network.org),[10] such as Standard Protocol Items: Recommendations for Interventional Trials (SPIRIT-AI),[11] Consolidated Standards of Reporting Trials (CONSORT-AI)[12] and Transparent Reporting of a Multivariable Prediction Model for Individual Prognosis or Diagnosis (TRIPOD-AI).[13]

STARD-AI is being coordinated by a global Project Team and Steering Committee consisting of clinician scientists, computer scientists, journal editors, EQUATOR Network representatives, reporting guideline developers, epidemiologists, statisticians, industry leaders, funders, health policy makers, legal experts and medical ethicists.

## Aim

This study aims to produce a specific reporting guideline (STARD-AI) for AI-centred diagnostic test accuracy studies.

## Focus of STARD-AI

The focus of STARD-AI is to aid the comprehensive reporting of research that use AI techniques to assess diagnostic test accuracy and performance. This can account for either single or combined test data, which often consists of either (1) imaging data (eg, CT scans), (2) pathological data (eg, digitised specimen slide) or (3) reporting data (eg, electronic health records). STARD-AI may also be used within studies which report on image segmentation and other relevant data classification techniques. If the emphasis of the study is on either developing, validating or updating a multivariable prediction model which produces an individualised probability of developing a condition (eg, time-to-event prediction), the TRIPOD-AI reporting guidelines may be more appropriate.

Typically, diagnostic test accuracy studies compare test results between participants who are either with or without a target condition. Data from study participants undergo assessment by an index test, which is designed to identify a specific target condition. This process occurs alongside a concurrent reference standard for the target condition within a defined time frame. Estimates of performance are typically based on a comparison between index test results and reference standard results from the same participant cohort. Alternatively, diagnostic performance can compare the performance of an index test against a reference standard determined through the incidence of an event within a defined time frame.

A significant number of contemporary AI diagnostic studies include information related to both the development and testing (validation) of AI-centred index tests. In order to accommodate and improve on this practice, STARD-AI will propose items related to AI index test development and validation as part of the consensus process. Other key topics for consideration within this study include, but are not limited to, the following: (1) data preprocessing methods, (2) AI index test development methods (eg, dataset partition, model calibration, stopping criteria when training, use of external validation sets), (3) fairness metrics, (5) non-standard performance metrics, (5) explainability and (6) human-AI index test interaction. As noted in the methods section, the inclusion of specific items related to these issues is reliant on consensus that is achieved through a transparent and fair evidence generation process.

*Sounderajah V, et al. BMJ Open* 2021;**11**:e047709. doi:10.1136/bmjopen-2020-047709

## METHODS

This protocol has been constructed in accordance with the EQUATOR Network toolkit for developing reporting guidelines.[14] It has also greatly benefitted from the experience and expertise from Project Team and Steering Committee members who had previously led the STARD 2003,[15] STARD 2015, STARD for Abstracts,[16] SPIRIT-AI and CONSORT-AI initiatives, respectively.

We can distil the development of the STARD-AI checklist into six stages. The overall goal of the STARD-AI initiative is to generate a list of minimally essential items, based on the established STARD 2015 framework, which should be reported in all AI diagnostic test accuracy studies. The items must assist the reader to appraise the completeness, applicability and potential for bias of the study findings.

### Stage 1: project organisation

A nine-member STARD-AI Project Team was established to coordinate the reporting guideline development process. The Project Team consists of the founder of STARD (PMB), the former UK Minister for Health and the current chair for the National Health Service Accelerated Access Collaborative (AD), members of the TRIPOD-AI core committee (GC and KM), a senior software engineer (SS), directors of the EQUATOR Network (DM and GC), the scientific content deputy editor for JAMA (RMG) as well as two clinician scientists from Imperial College London (HA and VS). The project team are responsible for identifying suitable members of the steering committee, candidate item generation, undertaking the online surveys for the modified Delphi consensus process, organising the consensus meeting, drafting the STARD-AI checklist and accompanying documents, piloting the draft STARD-AI checklist as well as leading on the dissemination process.

Further to the project team, a multidisciplinary STARD-AI Steering Committee was established to provide specialist guidance throughout. This committee consists of clinician scientists, computer scientists, journal editors, EQUATOR network directors, epidemiologists, statisticians, industry leaders, funders, health policy leaders, regulatory leaders, legal experts, patient representation experts and medical ethicists. These individuals were identified through their notable work with respect to (1) diagnostic accuracy research and its clinical translation, (2) applied AI in healthcare as well as (3) notable contribution to other AI-centred EQUATOR Network registered initiatives, such as TRIPOD-AI, CONSORT-AI and SPIRIT-AI.

Prior to stage 2, the STARD-AI project was registered with the EQUATOR Network.

### Stage 2: item generation

In order to generate a candidate list of items to enter the modified Delphi consensus process, the project team undertook a literature review, an online scoping survey with an international panel of experts and a patient public involvement and engagement exercise.

### Literature review

In January 2020, a literature review of both academic and non-academic literature was undertaken. An electronic database search of Medical Literature Analysis and Retrieval System Online and Excerpta Medica database (EMBASE) was conducted through Ovid. Both Medical Subject Headings or EMBASE Subject Headings (Emtree) were used. Search results were imported into Covidence (Covidence.org, Melbourne, Australia) for duplicate removal and study selection. Two individuals (VS and HA) individually screened study titles and abstracts for inclusion. Disagreements were resolved through discussion.

This process was augmented by non-systematic searches using grey literature, social networking platforms as well as personal article collections highlighted by members of the project team. Titles and abstracts of shortlisted publications were screened by one of two reviewers (VS and HA) and potentially eligible publications were retrieved for full-text assessment. Extracted material were broadly classified into four categories: (1) general considerations regarding diagnostic accuracy studies and AI, (2) evidence and statements suggesting modification to existing STARD 2015 items, (3) evidence and statements suggesting additions to the STARD 2015 checklist and (4) evidence and statements suggesting the removal of specific items from the STARD 2015 checklist.

### Online scoping survey

In addition to this, in February 2020, the project team undertook an online survey with an international panel of 80 experts in order to identify potential further items or modifications that warrant consideration. Written participant consent was attained as part of this process. This process generated over 2500 responses, which were analysed and classed into the aforementioned four broad categories.

### PPIE exercise

Lastly, a focus group was conducted with patients and members of the public who had expressed an interest in participating in forums related to digital health and AI. Written participant consent was attained as part of this process. The objective of these discussions was twofold: (1) to further identify issues not uncovered during the literature review and expert survey and (2) to gain further understanding of the perceived importance of specific items raised thus far. These discussions were conducted remotely using Zoom (Zoom Video Communications, USA).

An expert facilitator led a discussion on the current use of AI in healthcare, on what the aims of STARD-AI were and what participants considered to be important items to capture during the study process. As stakeholder discussions were conducted virtually on Zoom, anonymised post hoc discussion transcripts were maintained. Two investigators (VS and HA) independently identified common themes and subthemes from the discussion,

which were classed into the aforementioned four broad categories.

Having synthesised the findings of the literature review, the survey and the patient public involvement and engagement exercise, the project team, in collaboration with the steering committee, decided on which items warranted consideration in the formal modified Delphi consensus process.

## Stage 3: modified Delphi consensus process (ongoing)
### Study design and participants

This study has adopted a pragmatic modified Delphi consensus methodology. The Delphi consensus methodology is a well-established method[17] of obtaining a collective opinion from a group of experts through a series of questionnaires; each one refined based on feedback from respondents.

Participants were invited to join the STARD-AI Consensus Group on account of their expertise as clinician scientists, computer scientists, journal editors, EQUATOR Network representatives, reporting guideline developers, epidemiologists, statisticians, industry leaders (eg, clinician scientists, computer scientists and product managers from health technological companies), funders, health policy-makers, legal experts and medical ethicists. These experts were shortlisted through two principle means; either through the professional networks of members of the STARD-AI Project Team and Steering Committee or through recognition, critical involvement and achievements in a field that is related to diagnostic AI systems in the health sector (eg, authorship of seminal academic publications, key thought leaders, clinicians involved in prominent AI translational work and health policy directors, among others). Moreover, ensuring fair representation across geographies and demographics was a pertinent consideration during recruitment. Shortlisted participants were mutually agreed on by the project team members.

Following this, invited experts were provided with 3 weeks to respond to the initial invitation to participate. Written participant consent was attained as part of this process. Those who accepted the invitation were invited to complete each round of the modified Delphi consensus process. Those who contribute to both online rounds will be acknowledged by name as an author, within a group authorship model, in the publication that arises from this study.

In each round of the modified Delphi consensus process, participants are asked to grade each candidate item using a 5-point Likert-like scale (1—very important, 2—important, 3—moderately important, 4—slightly important, 5—not at all important). The threshold for consensus is predefined at ≥75%. Items which achieve ≥75% ratings of 1 or 2 are deemed to be essential for inclusion and are put forward for discussion in the final round (round 3, which will occur in the form of a virtual teleconference meeting). Items which achieve ≥75% ratings of 4 or 5 are deemed unimportant for inclusion and are

excluded. Items which do not reach this threshold of consensus are put forward to the next round of the modified Delphi consensus process. In addition to rating items, participants are asked in a free-text format to suggest any other items that they consider to be important to discuss in subsequent rounds.

In round 2, the survey will compose of (1) items for which consensus was not achieved in round 1 and (2) any new items suggested as part of round 1 feedback. Next to each item, participants will be reminded of what rating they gave in the previous round. Additionally, the mean score given by the overall group in the previous round will be displayed for each item. Thus, participants will be able to revise their initial score with the additional knowledge of peer responses. Following the collection of round 2 responses, additional items which achieve consensus as 'important' will be put forward for discussion during round 3. Those items that achieve consensus as 'unimportant' are excluded. Lastly, any non-consensus items from round 2 will be resolved through discussion among those in virtual attendance at the consensus meeting (round 3).

The consensus meeting (round 3) will consist of the STARD-AI Project Team and the STARD-AI Steering Committee. Given COVID-19 constraints, the meeting will be conducted virtually using Zoom. The primary objective is to develop a draft version of the STARD-AI checklist. As recommended in the Core Outcome Measures in Effectiveness Trials (COMET) handbook, the nominal group technique, a highly structured group interaction framework, will be utilised to aid this process.[18 19] Following a brief introduction and explanation of the purpose of the meeting by the facilitators (VS and HA), participants will discuss the inclusion and exclusion of candidate items. Participants will be asked to share any comments they have generated in a 'round robin' format until all contributions are exhausted. Participants will then be invited to discuss or seek further clarification about any of the ideas or comments produced. This discussion phase will be led by facilitators (VS and HA) to ensure that the discussion will not be dominated by any one individual and will be as neutral as possible.[20]

### Study conduct

VS and HA are the Delphi facilitators for the online survey rounds as well as the teleconference consensus meeting. They are responsible for the creation of the questionnaires, the invitations, the responses, the reminders, the analysis as well as the feedback for subsequent rounds.

The first two rounds of the modified Delphi consensus process are conducted as online surveys using the Delphi-Manager software (V.4.0), which is developed and maintained by the Core Outcome Measures in Effectiveness Trials initiative. Round 3 (the consensus meeting) will be carried using Zoom.

## Stage 4: development of the (1) checklist, (2) statement and (3) explanation and elaboration document

On completion of the modified Delphi consensus process, the project team will draft the initial STARD-AI checklist and statement. The draft checklist and statement will be shared among the wider steering committee in order to discuss its content and therefore allow the steering committee to suggest additions, subtractions or modifications as they see fit. This stage will also allow for harmonisation of key terms with the imminent TRIPOD-AI, in addition to the existing CONSORT-AI and SPIRIT-AI checklists.

## Stage 5: piloting phase

On completion of the first draft of the STARD-AI checklist, we intend to organise a piloting phase among expert users (Pilot Group). The main aim of these piloting sessions is to identify items which are considered to be vague, unnecessary or missing. We intend to undertake this process among radiology experts, pathology experts, computer scientists, expert statisticians, journal editorial boards, members of the global EQUATOR Network, key industry stakeholders as well as policy experts. Much like stage 3, these experts are shortlisted through two principle means; either through the professional networks of members of the STARD-AI Project Team and steering committee or through either (1) involvement in teams that have led diagnostic AI studies or (2) work as peer reviewers or editorial board members for journals that publish diagnostic AI studies. Experts are mutually agreed on by the Project Team members and Steering Committee. Feedback will be captured through surveys and a series of semi-structured interviews. This approach allows for the capture of broad issues through surveys, which form themes that can be further explored in detail during semistructured interviews. Anonymised feedback from the interviews will be transcribed to allow for thematic analysis so that recurring trends are appropriately identified and presented back to the project team and steering committee for discussion. Experts within the Pilot Group will be acknowledged by name as an author, within a group authorship model, in the publications that arise from this study.

In conjunction to this piloting process, the project team will also prepare the explanation and elaboration (E&E) document to provide rationale for the included items alongside examples of good reporting.

## Stage 6: finalisation, publication and postpublication activities

Following the piloting phase, the final proposed amendments to STARD-AI will be discussed among the project team and the steering committee. Once consensus has been reached through email correspondence, the checklist and accompanying documents will be disseminated.

The dissemination strategy will be principally tailored towards five groups of stakeholders: (1) academia, (2) policy, (3) guidelines and regulation, (4) industry and (5) patient representing bodies. Although a significant amount of material will cross over between stakeholders, creating specific material is considered to be the most meaningful way of achieving impact.

We aim to publish the STARD-AI checklist, the accompanying statement and the E&E document in an open access format (through a CC-BY licence). In order to further complement this, we aim to create specialty-specific discourse regarding STARD-AI through focused editorials in pertinent journals. These journal editors will also be actively encouraged to endorse STARD-AI as part of their broader editorial policy. Moreover, we will present STARD-AI at national and international scientific meetings. Translations of the guideline in various languages are actively encouraged (available on the EQUATOR network) in order to further broaden the scope of its impact. We encourage interested parties to contact the corresponding author for further information about the translation policies.

In addition to this, we aim to persuade governmental bodies to adopt the checklist as part of their policy assessments. This will involve presentations at national and international health policy summits (eg, World Innovation Summit for Health and National Health Service (NHS) Accelerated Access Collaborative meetings). Furthermore, we will aim to integrate teaching about STARD-AI into national health policy educational programmes through pre-existing collaborations with academic institutions, NHS Digital Academy and NHSX.

Concurrent to this workstream will be our work with guidelines and regulatory bodies so that they may account for STARD-AI as part of their national health technology assessments. This will involve the US Food and Drug Administration, the Medicines and Healthcare products Regulatory Agency and The National Institute for Health and Care Excellence among others.

Lastly, we will present STARD-AI to a broad range of health technology companies so that their product pipelines may accommodate for this downstream mode of assessment.

## CONCLUSION

STARD-AI will serve as the first global consensus achieved guidance for the reporting of AI centred diagnostic accuracy studies. Through a clear multistakeholder dissemination policy, we hope that STARD-AI can significantly contribute towards minimising research waste as well as serving as an instrument that assists the streamlined translation of these nascent technologies. We anticipate that STARD-AI will be published in Q3 2021.

## Ethics

Ethical approval has been granted by the Joint Research Compliance Office at Imperial College London (SETREC reference number: 19IC5679).

## GLOSSARY

### Project team
This consists of the founder of STARD (PMB), the former UK Minister for Health and the current chair for the NHS Accelerated Access Collaborative (AKD), members of the TRIPOD-AI group (GC and KM), a senior software engineer (SS), directors of the EQUATOR Network (DM and GC), the scientific content deputy editor for JAMA (RMG) as well as two clinician scientists from Imperial College London (HA and VS).

### Steering committee
This consists of clinician scientists, computer scientists, journal editors, EQUATOR Network representatives, epidemiologists, statisticians, industry leaders, funders, health policy-makers, legal experts and medical ethicists.

### Consensus group
This consists of experts who participated in the modified Delphi consensus process (stage 3) of the study.

### Pilot group
This consists of experts who participated in the pilot phase (stage 5) of the study.

### Checklist
A document listing the minimally essential items that should be reported in all diagnostic test accuracy studies centred around AI-centred index tests. This constitutes the core of the reporting guideline.

### Statement
A document which provides the rationale underpinning the reporting guideline and describes the process of developing the associated documents.

### Explanation and elaboration
A document which provides the rationale behind each item in the checklist alongside examples of good reporting.

### Reporting guideline
The combination of the checklist, statement and E&E documents.

### Artificial Intelligence
The science of developing computer systems which can perform tasks which normally require human intelligence.

### Modified Delphi study
A research method that derives the collective opinions of a group through a staged consultation of surveys, questionnaires or interviews, with an aim to reach consensus at the end.

### Author affiliations
[1]Department of Surgery and Cancer, Imperial College London, Paddington, UK
[2]Institute of Global Health Innovation, Imperial College London, London, UK
[3]Journal of the American Medical Association, Chicago, Illinois, USA
[4]Google Health, Pal Alto, California, USA
[5]DeepMind Technologies Ltd, London, UK
[6]Cochrane Netherlands, University Medical Center Utrecht, University of Utrecht, Utrecht, The Netherlands
[7]Department of Epidemiology, Julius Center for Health Sciences and Primary Care, Utrecht, The Netherlands
[8]Centre for Statistics in Medicine, Nuffield Department of Orthopaedics, Rheumatology and Musculoskeletal Sciences, University of Oxford, Oxford, UK
[9]Centre for Journalology, Ottawa Hospital Research Institute, Ottawa, Ontario, Canada
[10]Department of Epidemiology and Data Science, Amsterdam University Medical Centres, Duivendrecht, The Netherlands
[11]Google Health, London, UK
[12]Institute of Inflammation and Ageing, College of Medical and Dental Sciences, University of Birmingham, Birmingham, UK
[13]University Hospitals Birmingham NHS Foundation Trust, Birmingham, UK
[14]Health Data Research UK, London, UK
[15]Birmingham Health Partners Centre for Regulatory Science and Innovation, University of Birmingham, Birmingham, UK
[16]The Alan Turing Institute, London, UK
[17]Singapore Eye Research Institute, Singapore National Eye Center, Singapore
[18]University of Leeds, Leeds, UK
[19]Optum, Paddington, London, UK
[20]Department of Primary Care and Public Health, Imperial College London, London, UK
[21]School of Public Health, Imperial College London, London, UK
[22]Department of Radiology, University of Ottawa, Ottawa, Ontario, Canada
[23]Harvard Medical School, Boston, Massachusetts, USA
[24]School of Social and Community Medicine, University of Bristol, Bristol, UK
[25]Division of Health Policy and Management, Harvard T.H. Chan School of Public Health, Boston, Massachusetts, USA

**Contributors** VS, HA, RMG, SS, JDF, LH, KM, GC, DM, PMB and AD were involved in the planning and design of the study. VS drafted the manuscript with all authors contributing to edits. AK, AKD, BAM, DTi, DTr, DK, FG, JG, JP-S, LH, MM, NR, NT, PN, PW, RA, SV, SRM, TP and XL are members of the STARD-AI Steering Committee. They are equally involved in the wider conduct and direction of the overall study. All of the authors edited the manuscript and provided critical appraisal. All named authors approved the final draft of the manuscript.

**Funding** Infrastructure support for this research was provided by the NIHR Imperial Biomedical Research Centre (BRC). GC is supported by the NIHR Oxford Biomedical Research Centre and Cancer Research UK (programme grant: C49297/A27294). DT is funded by National Pathology Imaging Co-operative, NPIC (Project no. 104687) is supported by a £50m investment from the Data to Early Diagnosis and Precision Medicine strand of the government's Industrial Strategy Challenge Fund, managed and delivered by UK Research and Innovation (UKRI). FG is supported by the National Institute for Health Research Applied Research Collaboration Northwest London.

**Competing interests** None declared.

**Patient and public involvement** Patients and/or the public were involved in the design, or conduct, or reporting, or dissemination plans of this research. Refer to the Methods section for further details.

**Patient consent for publication** Not required.

**Provenance and peer review** Not commissioned; externally peer reviewed.

**ORCID iDs**
Viknesh Sounderajah http://orcid.org/0000-0002-4595-8402
Gary Collins http://orcid.org/0000-0002-2772-2316
David Moher http://orcid.org/0000-0003-2434-4206
Bilal Akhter Mateen http://orcid.org/0000-0003-4423-6472
Sheraz R Markar http://orcid.org/0000-0001-8650-2017

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
