## [Reviewer comments · BMJ Open]

ARTICLE DETAILS

TITLE (PROVISIONAL)	Developing a Reporting Guideline for Artificial Intelligence Centred Diagnostic Test Accuracy Studies: The STARD-AI Protocol
AUTHORS	Sounderajah, Viknesh; Ashrafian, Hutun; Golub, Robert; Shetty, Shravya; De Fauw, Jeffrey; Hooft, Lotty; Moons, Karel; Collins, Gary; Moher, David; Bossuyt, Patrick M; Darzi, Ara; Karthikesalingam, Alan; Denniston, Alastair; Mateen, Bilal Akhter; Ting, Daniel; Treanor, Darren; King, Dominic; Greaves, Felix; Godwin, Jonathan; Pearson-Stuttard, Jonathan; Harling, Leanne; McInnes, Matthew; Rifai, Nader; Tomasev, Nenad; Normahani, Pasha; Whiting, Penny; Aggarwal, Ravi; Vollmer, Sebastian; Markar, Sheraz; Panch, Trishan; Liu, Xiaoxuan

VERSION 1 – REVIEW

REVIEWER	Seong Ho Park University of Ulsan
REVIEW RETURNED	30-Jan-2021

GENERAL COMMENTS	This manuscript is a well-written summary that explains how the STARD-AI will be put together. The potential importance of reporting guidelines for AI-related research studies in healthcare such as the STARD-AI cannot be overstated. Therefore, I applaud the efforts of all of those who are involved in this meaningful work. As the entire project has been progressing for some time (currently, stages 1 and 2 have been completed), any suggestions in retrospect regarding the previous steps would not be practical. Therefore, I have some comments regarding the conduct of stages 3 and 5 and some other tips for improved clarity of this protocol article and the final STARD-AI guidelines. 1. It is a bit unclear how the participants for stage 3 (the STARD-AI Consensus Group) will be recruited. General explanations are present. However, a more specific description of the plan would be beneficial.2. Similarly, it is obscure how they will recruit the experts and non-experts for stage 5 (the piloting stage). Also, unlike the well-established modified Delphi process in stage 3, it is unclear how they will analyze and process the feedback from stage 5 in adjusting the guidelines.3. I am glad to see that the STARD-AI will account for image segmentation specifically. Similarly, further clarification on the scope of the STARD-AI, what it covers and what it doesn't, would help the users. Below are some points worth considering. 1) The distinction from TRIPOD-AI by the "multivariable prediction vs. not" seems a bit confusing. AI itself, whether deep learning or
---

	conventional machine learning, is high-dimensional modeling. Therefore, a distinction by "multivariable vs. not" does not seem fit. The STARD-AI seems more relevant for the evaluation of the accuracy/performance of an end product. 2) Even from evaluating the accuracy/performance of an end product, some obscurity remains. Many diagnostic AI algorithms present "probability." Therefore, the diagnostic accuracy (for discrimination) is often accompanied by calibration accuracy in many AI-related studies. Sometimes, discrimination and calibration accuracies are a bit inseparable. Maybe, this issue could be addressed from the threshold (for ROC) viewpoint. Calibration is mentioned in TRIPOD, but not in STARD. How would the STARD-AI address this? 3) Given that practically all AI models are multivariable, some distinction by the nature of diagnostic/predictive tasks would be more practical. TRIPOD matches time-to-event ("survival") prediction better from a user perspective, whereas STARD is for static binary discrimination. I wonder how the STARD-AI group has been communicating with the TRIPOD-AI group regarding the differences in their guidelines' scopes. 4) I wonder if the STARD-AI would address more specifically the studies that include (alternative) free-response ROC analysis. The original STARD does not explicitly address it. Due to the technical prowess of current AI methods, studies on CAD-type AI models are a lot more frequently reported than in the past. Many of such studies include (alternative) free-response ROC analysis. Given the increase in such studies in the field of AI, the STARD-AI might need to address it more specifically. 5) Another critical component in AI-related diagnostic/predictive studies is the methods and results on explainability (or interpretability). Currently, some kinds of activation maps are typically used. This issue does not exist in the original STARD. It may need to be clarified whether the STARD-AI would specifically address this issue.
--	--

REVIEWER	Helen Storey PATH, Diagnostics
REVIEW RETURNED	23-Feb-2021

GENERAL COMMENTS	Thank you to the authors for their effort. This article is well written and relevant for improving future reporting of studies evaluating the diagnostic accuracy of AI techniques. A few minor suggestions for consideration. 1) Consistent use of future or past tense in the methods section would be helpful for readability. (example: page 16, line 51) 2) Please double check for typos (some examples though not all: line 3 page 20, line 38 page 22) 3) Consider condensing the dissemination section (stage 6). Detail by stakeholder group is a bit more protracted than needed to convey the effort. 4) Some sort of concluding remark would be useful to remind readers why this work is being done. What do you hope is achieved by creation of STARD-AI? When will it be completed?
--

VERSION 1 – AUTHOR RESPONSE

Reviewer: 1

Dr. Seong Ho Park, University of Ulsan

Comments to the Author:

This manuscript is a well-written summary that explains how the STARD-AI will be put together. The potential importance of reporting guidelines for AI-related research studies in healthcare such as the STARD-AI cannot be overstated. Therefore, I applaud the efforts of all of those who are involved in this meaningful work.

As the entire project has been progressing for some time (currently, stages 1 and 2 have been completed), any suggestions in retrospect regarding the previous steps would not be practical. Therefore, I have some comments regarding the conduct of stages 3 and 5 and some other tips for improved clarity of this protocol article and the final STARD-AI guidelines.

1. It is a bit unclear how the participants for stage 3 (the STARD-AI Consensus Group) will be recruited. General explanations are present. However, a more specific description of the plan would be beneficial.

Thank you for highlighting this - we have added further detail to this section (line 336 of the manuscript document).

2. Similarly, it is obscure how they will recruit the experts and non-experts for stage 5 (the piloting stage). Also, unlike the well-established modified Delphi process in stage 3, it is unclear how they will analyze and process the feedback from stage 5 in adjusting the guidelines.

Thank you for highlighting this - we have added further detail to this section (line 411 of the manuscript document).

3. I am glad to see that the STARD-AI will account for image segmentation specifically. Similarly, further clarification on the scope of the STARD-AI, what it covers and what it doesn't, would help the users. Below are some points worth considering.

1) The distinction from TRIPOD-AI by the "multivariable prediction vs. not" seems a bit confusing. AI itself, whether deep learning or conventional machine learning, is high-dimensional modeling. Therefore, a distinction by "multivariable vs. not" does not seem fit. The STARD-AI seems more relevant for the evaluation of the accuracy/performance of an end product.

Thank you for highlighting this - this is a very fair point and we have amended our 'Focus of STARD-AI' section to reflect a more accurate representation of our mission statement (line 217 of the manuscript document onwards). STARD-AI will concentrate on the reporting of diagnostic accuracy/performance from single or combined medical tests. TRIPOD-AI will concentrate primarily upon the development and validation of prognostic models from multiple datasets.

2) Even from evaluating the accuracy/performance of an end product, some obscurity remains. Many diagnostic AI algorithms present "probability." Therefore, the diagnostic accuracy (for discrimination)

is often accompanied by calibration accuracy in many AI-related studies. Sometimes, discrimination and calibration accuracies are a bit inseparable. Maybe, this issue could be addressed from the threshold (for ROC) viewpoint. Calibration is mentioned in TRIPOD, but not in STARD. How would the STARD-AI address this?

We completely agree with you. Since we submitted this paper in December, we have started our Delphi consensus process. As part of the Round 1 results, calibration has been noted as something that STARD-AI should incorporate as part of a wider model development section (the item related to calibration has achieved consensus and will be discussed at the round 3 meeting in due course). This has been highlighted in line 240 onwards.

3) Given that practically all AI models are multivariable, some distinction by the nature of diagnostic/predictive tasks would be more practical. TRIPOD matches time-to-event ("survival") prediction better from a user perspective, whereas STARD is for static binary discrimination. I wonder how the STARD-AI group has been communicating with the TRIPOD-AI group regarding the differences in their guidelines' scopes.

Thank you for this very apt and timely comment! Two members of our steering committee are also TRIPOD-AI steering committee members and we have recently spent some time delineating our respective scopes for ease of end-user adoption. We hope that this is more clearly reflected in line 217 onwards.

4) I wonder if the STARD-AI would address more specifically the studies that include (alternative) free-response ROC analysis. The original STARD does not explicitly address it. Due to the technical prowess of current AI methods, studies on CAD-type AI models are a lot more frequently reported than in the past. Many of such studies include (alternative) free-response ROC analysis. Given the increase in such studies in the field of AI, the STARD-AI might need to address it more specifically.

Thank you for highlighting this pertinent point around performance metrics. This has been highlighted in line 240. A specific item regarding performance metrics that are either 'non-standard'/emerging/those not covered in STARD 2015 has reached consensus in round 1 of our modified Delphi consensus process and will be discussed at the round 3 meeting for potential inclusion in the STARD-AI checklist.

5) Another critical component in AI-related diagnostic/predictive studies is the methods and results on explainability (or interpretability). Currently, some kinds of activation maps are typically used. This issue does not exist in the original STARD. It may need to be clarified whether the STARD-AI would specifically address this issue.

Thank you for highlighting this pertinent point around 'explainability'. This has been highlighted in line 240. A specific item regarding 'explainability' (with the potential inclusion of class activation maps as a case example) has reached consensus and will be discussed at the round 3 meeting for potential inclusion in the STARD-AI checklist.

N.B.

All of these issues that you have kindly noted have featured prominently in our evidence generation process and will mostly likely feature in some form in our eventual checklist and explanation & elaboration document. We are slightly wary of making firm promises as the precise nature of their inclusion is yet to be finalised. I hope that our paragraph from line 244 demonstrates that these are key considerations for us in this process without assuming results from the consensus process.

Reviewer: 2
Mrs. Helen Storey, PATH

Comments to the Author:

Thank you to the authors for their effort. This article is well written and relevant for improving future reporting of studies evaluating the diagnostic accuracy of AI techniques. A few minor suggestions for consideration.

1) Consistent use of future or past tense in the methods section would be helpful for readability. (example: page 16, line 51)

Apologies for these inconsistencies. We have endeavoured to correct these issues across the manuscript (including the example that you have kindly highlighted). The deliberate switch between past and future occurs at Stage 3, which is where the study group currently are.

2) Please double check for typos (some examples though not all: line 3 page 20, line 38 page 22)

Apologies for these inconsistencies. We have endeavoured to correct these issues across the manuscript (including the example that you have kindly highlighted)

3) Consider condensing the dissemination section (stage 6). Detail by stakeholder group is a bit more protracted than needed to convey the effort.

Thank you for highlighting this – we have condensed this entire section (line 427 of manuscript document onwards), as per recommendations.

4) Some sort of concluding remark would be useful to remind readers why this work is being done. What do you hope is achieved by creation of STARD-AI? When will it be completed?

Thank you for highlighting this – we have added a brief conclusion (line 456 of manuscript document onwards), which we hope provides a suitable concluding remark regarding the scope of the project, its potential translational benefit as well as the estimated completion date.

VERSION 2 – REVIEW

REVIEWER	Seong Ho Park University of Ulsan
REVIEW RETURNED	16-May-2021
GENERAL COMMENTS	The authors have adequately addressed the comments of the reviewers. Thank you for the efforts.